# Synergistic Anticancer Effects of Fibroblast Growth Factor Receptor Inhibitor and Cannabidiol in Colorectal Cancer

**DOI:** 10.3390/nu17162609

**Published:** 2025-08-12

**Authors:** Yeonuk Ju, Bu Gyeom Kim, Jeong-An Gim, Jun Woo Bong, Chin Ock Cheong, Sang Cheul Oh, Sang Hee Kang, Byung Wook Min, Sun Il Lee

**Affiliations:** 1Division of Colon and Rectal Surgery, Department of Surgery, Korea University Guro Hospital, Korea University College of Medicine, Seoul 08308, Republic of Korea; snrlsnrl@korea.ac.kr (Y.J.); zest815@gmail.com (J.W.B.); owho9@naver.com (C.O.C.); kasaha1@korea.ac.kr (S.H.K.); gsmin@korea.ac.kr (B.W.M.); 2Institute of Convergence New Drug Development, Korea University College of Medicine, Seoul 08308, Republic of Korea; qnrua10047@korea.ac.kr; 3Department of Medical Science, Soonchunhyang University, Seoul 31538, Republic of Korea; vitastar@sch.ac.kr; 4Division of Oncology, Department of Internal Medicine, Korea University Guro Hospital, Korea University College of Medicine, Seoul 08308, Republic of Korea; sachoh@korea.ac.kr

**Keywords:** colorectal cancer, FGFR, CBD, DDIT3/CHOP, ER stress

## Abstract

Background/Objectives: Colorectal cancer (CRC) remains a significant global health concern, with limited treatment options for metastatic stage 4 CRC. Fibroblast Growth Factor Receptor (FGFR) is a promising therapeutic target in CRC, while cannabidiol (CBD) has shown potential for inducing cell death and overcoming drug resistance. This study evaluates the efficacy of FGFR inhibitors and explores the synergistic effects of combining FGFR inhibitors with CBD in inducing apoptosis in CRC cells. Methods: Cannabidiol and FGFR inhibitors were applied, and protein expression was analyzed via Western blot. Cell viability was assessed using the WST-1 assay, while apoptosis was measured through flow cytometry using Annexin V-FITC/PI staining. CHOP-specific siRNA transfection was performed to study gene silencing effects, followed by RNA sequencing for differential expression and pathway analysis. Statistical significance was determined using ANOVA and *t*-tests, with *p* < 0.05. Results: FGFR expression patterns were confirmed in various cancer cell lines, with NCI-H716 showing high FGFR2 expression. Treatment with CBD (4 µM) and AZD4547 (10 nM) resulted in significant cell death, especially when used in combination, indicating the effectiveness of this combined therapy. Increased apoptosis in NCI-H716 cells was confirmed with the combined treatment. RNA sequencing and heatmap analysis suggested that ER stress might be related to the observed synergistic effect. The role of ER stress in the combination-induced apoptosis of NCI-H716 cells was further validated. Conclusions: The combination of FGFR inhibitors and cannabidiol exhibited a synergistic effect in inducing cell death in colorectal cancer cells, likely through the ER stress pathway. This study supports the potential of combined FGFR inhibitor and CBD therapy as a promising strategy for enhancing anticancer effects in CRC.

## 1. Introduction

Colorectal cancer (CRC) is the third most common cancer worldwide and continues to be a major global health concern. Treatment typically involves a combination of three traditional oncologic strategies: surgery, radiation therapy, and chemotherapy. In cases without metastasis, the five-year survival rate can surpass 70–90%. However, this rate significantly drops to 15–20% for stage IV disease [1,2]. Recent advances in treating stage IV metastatic CRC have led to the development of integrated therapies that combine surgery, chemotherapeutic agents, and targeted anticancer drugs. These multidisciplinary approaches have significantly improved five-year survival rates to above 50% [3,4,5]. In instances of unresectable, progressive, or recurrent CRC, the advent of targeted anticancer drugs has facilitated the application of combined multidisciplinary treatments, aiming to reduce tumor size and extend patient survival. Despite these advancements, the variety of available treatment options remains limited, and there is an urgent need for the development of effective therapeutic strategies for recurrent and drug-resistant CRC [6,7].

The Fibroblast Growth Factor (FGF) and its receptors (FGFRs) constitute a signaling pathway critical for cell development, differentiation, survival, migration, angiogenesis, and carcinogenesis. This pathway involves four receptors and 22 ligands, which are activated through ligand-induced dimerization and phosphorylation, triggering several downstream pathways, including RAS–RAF–MEK–MAPK, PI3K–AKT–mTOR, JAK–STAT, and PLCγ [8]. Aberrant activation of FGFR signaling contributes to cancer cell proliferation, apoptosis resistance, invasion, and metastasis [9,10,11]. Several FGFR-targeted therapies are under investigation, and some have been approved for malignancies such as bladder cancer and cholangiocarcinoma [9]. However, their clinical benefit in CRC has been modest, partly due to the relatively low prevalence of FGFR alterations (≈4–5% in CRC), the emergence of acquired resistance, and class-specific toxicities such as hyperphosphatemia, fatigue, and gastrointestinal adverse effects [9,12]. These limitations highlight the need for novel strategies to enhance the efficacy of FGFR-targeted therapy. FGFR2 alterations in cancer can occur through various mechanisms, including gene amplification, point mutations, or protein overexpression [9,10,11,12]. While amplification and overexpression may be related, they are not synonymous and can have different biological and clinical implications. In the present study, we specifically focused on FGFR2 protein overexpression, particularly in the NCI-H716 colorectal cancer cell line, as determined by Western blotting, without direct assessment of gene amplification.

Cannabidiol (CBD) is a major non-psychotomimetic phytocannabinoid derived from Cannabis sativa that is under investigation as a potential therapeutic agent for inflammatory, neurodegenerative, and malignant diseases [13]. CBD has demonstrated anticancer activity in various malignancies, including breast cancer, glioma, prostate cancer, and CRC [14,15,16,17]. In CRC cells, CBD has been shown to induce cell death through multiple mechanisms, including reactive oxygen species (ROS) generation, mitochondrial dysfunction, MAPK pathway activation, and ER stress induction [15,16,17]. Notably, CBD activates MAPK signaling in CRC cells [17], which is a principal downstream effector pathway of FGFR. This overlap suggests potential crosstalk between CBD-induced signaling and FGFR-mediated pathways. Moreover, some evidence suggests that FGFR signaling may attenuate ER stress-mediated apoptosis [18], raising the possibility that FGFR inhibition could amplify CBD-driven ER stress and apoptotic responses.

Based on these mechanistic intersections, we hypothesized that FGFR inhibition may enhance the pro-apoptotic effects of CBD in CRC cells, particularly in tumors with high FGFR2 expression. This study aimed to evaluate the therapeutic potential of combining FGFR inhibitors with CBD, investigate the underlying molecular mechanisms—focusing on ER stress pathways—and assess whether this combination can elicit a synergistic antitumor effect in FGFR2-overexpressing CRC cells.

## 2. Materials and Methods

### 2.1. Cell Culture

This is an in vitro study using established human-derived colorectal cancer cell lines; no animal experiments or patient-derived live samples were involved. Cell lines utilized in this study included normal colon cells (CCD-18Co) and a variety of colon cancer cell lines (HCT116, NCI-H716, HT-29, Colo205, DLD-1, SW480, SW620), along with gastric cancer cell lines (MKN45, SNU16, KATOIII). These cells were obtained from the American Type Culture Collection (ATCC, Manassas, VA, USA) and the Korea Cell Line Bank (KCLB, Seoul, Republic of Korea). All cell lines were authenticated and confirmed to be free of mycoplasma contamination prior to use. Cells were cultured in RPMI 1640 medium (Gibco, Grand Island, NY, USA) supplemented with 10% fetal bovine serum (FBS; Gibco) and 100 mg/mL penicillin/streptomycin (P/S; GenDEPOT, Barker, TX, USA), and maintained under standard conditions (37 °C, 5% CO_2_, 95% humidity). All cell culture work was performed by trained laboratory personnel. The study protocol was approved by the Institutional Review Board of Korea University Guro Hospital (IRB No. 2022GR0449), and informed consent was waived due to the use of archived and anonymized cell lines.

### 2.2. Reagents and Antibodies

Cannabidiol was purchased from Hammer Enterprises (Evergreen, CO, USA) and stored at 4 °C until use. FGFR inhibitors (AZD4547, PD173074, BGJ398) were purchased from Selleckchem (Houston, TX, USA). The antibodies used and their sources were as follows: FGFR2, p-FGFR, AKT, p-AKT, ERK, p-ERK, stat3, p-stat3, Cleaved Poly (ADP-ribose) polymerase (PARP), Cleaved Caspase-3, Cleaved Caspase-8, Cleaved Caspase-9, eIF2α, phospho-eIF2α, Activating transcription factor 4 (ATF4), ATF3, inositol requiring enzyme-1α (IRE1α), and p-IRE1α were purchased from Cell Signaling Technology (Danvers, MA, USA). FGFR1, FGFR3, FGFR4, Glucose Regulated Protein 94 (GRP94), Bip, XBP-1s, and CHOP were purchased from Santa Cruz Biotechnology (Dallas, TX, USA). Anti-β-Actin was purchased from Sigma Aldrich (St. Louis, MO, USA).

### 2.3. Small Interfering RNA (siRNA)

CHOP siRNA, and control siRNA were purchased from Santa Cruz Biotechnology (Dallas, TX, USA). The cells were transfected with siRNA oligonucleotides using the Lipofectamine RNA iMAX reagent (Invitrogen, Carlsbad, CA, USA) according to the manufacturer’s instructions.

### 2.4. WST-1 Assay

Colorectal cancer cell lines were seeded at a density of 1 × 104 cells per 200 µL in each well and treated with varying concentrations of FGFR inhibitors and CBD, either alone or in combination. After 24 h of treatment, cell viability was assessed using a Cell Viability Assay Kit (EZ-Cytox, DOGEN, Daejeon, Republic of Korea). To this end, 10 µL of the EZ-Cytox reagent was added to each well, followed by a 2 h incubation at 37 °C. The absorbance was then measured at a wavelength of 450 nm to determine cell viability, which was calculated relative to a control group considered 100% viable.

### 2.5. Western Blotting

Cells were lysed in RIPA buffer (50 mM Tris, 150 mM NaCl, 1% Triton X-100 (Sigma-Aldrich, St. Louis, MO, USA), 0.1% SDS, and 1% sodium deoxycholate, pH 7.4), supplemented with protease and phosphatase inhibitors from Sigma Aldrich. Protein levels were quantified using a bicinchoninic acid assay from Thermo Fisher Scientific (Waltham, MA, USA). Proteins were then resolved by SDS-PAGE and electroblotted onto nitrocellulose membranes from GE Healthcare Life Sciences (Little Chalfont, UK). Membranes were blocked in TBS containing 0.2% Tween 20 and 5% skim milk, followed by overnight incubation at 4 °C with antibodies. Protein signals were visualized using X-ray film to assess the expression and modification of the target proteins. Blots were normalized to β-actin as a loading control. All Western blot experiments were independently repeated at least three times, yielding consistent results. The blots shown in the figures are representative images from these independent experiments.

### 2.6. Morphological Transformation Analysis

The impact of treatments on cell death was assessed through an analysis of morphological changes. Cells were plated in 50 µL volumes in 60Ø cell culture dishes and allowed to incubate for 24 h. Subsequently, they were treated with CBD, AZD4547, and a combination of both agents for an additional 24 h period. Post-treatment, cells were detached using Trypsin-EDTA (GenDEPOT, Barker, TX, USA) and then reseeded into a 6-well plate. Over a period of two weeks, the cells were observed for any morphological alterations and the extent of cell death was documented.

### 2.7. Flow Cytometry Analysis of Cell Apoptosis

To assess apoptosis, phosphatidylserine translocation was analyzed using both FITC and allophycocyanin-conjugated Annexin V. NCI-H716 cells, treated with cannabidiol, AZD4547, or both, were harvested and mixed with Annexin V and phosphatidylinositol (PI) reagent from the Annexin V-FITC Apoptosis Detection Kit (BioBud, Seoul, Republic of Korea). NCI-H716 cells were trypsinized and centrifuged at 1200 rpm for 5 min. The cell pellet was resuspended and stained with annexin V-FITC/PI solution. After a 30 min incubation at 4 °C or room temperature (RT) in the dark, cells were immediately analyzed by flow cytometry (BD LSR Fortessa-X20, BD Biosciences, San Joes, CA, USA) to detect apoptotic changes. Apoptotic cell death was quantified using FlowJo software v10.10 (BD Biosciences).

### 2.8. RNA Interference Assay

NCI-H716 cells underwent transfection with CHOP-specific siRNA to investigate the gene silencing effects on cellular response. Prior to the introduction of siRNA, cells were prepared by adding optimal minimal essential medium (Opti-MEM; Gibco) to the culture dish, which was then incubated at 37 °C for 30 min to achieve temperature equilibration. Subsequently, a transfection mixture containing siRNA, Opti-MEM, and Lipofectamine RNAiMAX was prepared and allowed to form siRNA-lipid complexes by incubating at room temperature for 30 min. This transfection complex was then gently added to the pre-warmed cells, followed by an incubation period of 18 h at 37 °C in a 5% CO_2_ incubator to facilitate cellular uptake. Post-transfection, cells were subjected to treatment with CBD and AZD4547 to enable downstream analyses of the effects of gene silencing in combination with these agents.

### 2.9. RNA Sequencing

RNA sequencing was performed on a set of NCI-H716 colon cancer cells, which included untreated controls and groups subjected to treatments with an FGFR inhibitor (AZD4547), CBD, and their combination. The total RNA was meticulously extracted using TRIzol reagent (Thermo Fisher Scientific), in accordance with the manufacturer’s specifications. Ensuring the highest standard, only RNA with a RIN score of 7 or greater was deemed suitable for RNA-Seq library construction. The sequencing process was expertly executed by Macrogen Inc. (Seoul, Republic of Korea).

Following sequencing, the data were processed for enrichment analysis to identify gene ontologies and pathways potentially impacted by the treatments. To visualize differential expression, we created heatmaps to depict patterns across conditions and box plots to display the expression levels of specific genes within each group. The enrichment of biological pathways was determined by KEGG analysis, while gene ontology, gene concept networks, and functional classification were performed with the Cluster Profiler and DOSE R packages, with a *p*-value threshold of <0.05 indicating statistical significance.

### 2.10. Statistical Analysis

For each experimental condition, a minimum of three independent replicates were conducted to ensure the robustness of the findings. Statistical analysis was performed using GraphPad Prism version 8.0.2 (GraphPad Software, San Diego, CA, USA). One-way Analysis of Variance (ANOVA) followed by Tukey’s post hoc test was applied for comparisons across multiple groups. For pairwise comparisons, the unpaired *t*-test was utilized. A *p*-value threshold of less than 0.05 (*p* < 0.05) was established to denote statistically significant differences.

## 3. Results

### 3.1. FGFR2 Expression and Response to CBD and FGFR Inhibitor in Colorectal Cancer Cells

Our initial studies examined FGFR family expression patterns across various cancer cell lines, focusing on colorectal cancer (CRC). Western blot analysis revealed that the NCI-H716 cell line expressed high levels of FGFR2 compared to the other CRC and non-CRC lines (Figure 1A). Treatment of CRC cell lines with AZD4547 or CBD for 24 h demonstrated variable sensitivity, with NCI-H716 cells showing a marked reduction in viability (Figure 1B). Dose–response analyses determined the optimal concentrations for the subsequent experiments as 4 μM for CBD (Figure 1C) and 10 nM for AZD4547 (Figure 1C). Treatment of NCI-H716 cells with AZD4547 alone inhibited FGFR signaling, as confirmed by Western blotting (Figure 1D). Other FGFR inhibitors, including PD173074 and BGJ398, produced similar inhibitory effects on NCI-H716 cell proliferation and FGFR downstream signaling (Appendix A).

### 3.2. Cytotoxic Assay of CBD and FGFR Inhibitor in Colorectal Cancer Cells

Combined treatment of NCI-H716 cells with CBD and AZD4547 produced marked morphological changes indicative of cytotoxicity (Figure 2A). Western blotting showed strong upregulation of apoptosis-related proteins (cleaved PARP, cleaved caspase-3, cleaved caspase-8) in FGFR2-high NCI-H716 cells, but not in FGFR-low CRC lines such as HCT116, DLD-1, and HT29 (Appendix A). When CBD was combined with other FGFR inhibitors (PD173074 or BGJ398), apoptotic protein induction was also enhanced in NCI-H716 cells (Appendix A). Cell viability assays confirmed that combination therapy significantly reduced proliferation compared to single-agent treatments (Figure 2C).

### 3.3. Enhanced Apoptotic Response from Combined Treatment

Flow cytometry analysis with Annexin V/PI staining demonstrated that the CBD + AZD4547 combination significantly increased apoptotic cell populations compared to either treatment alone (Figure 3A). Quantitative analysis confirmed a statistically significant synergistic effect in inducing apoptosis in NCI-H716 cells (Figure 3B).

### 3.4. RNA Sequencing and Analysis Insights to Examine Mechanisms Underlying Combination Synergy in Therapy

RNA sequencing identified differentially expressed genes (DEGs) following single and combination treatments. Volcano plots and box plots showed that ER stress-related genes, including ATF3 and DDIT3 (CHOP) from gene-level analysis and DDIT3.3 from transcript-level analysis, were significantly upregulated with combination therapy (Figure 4A,B). Additional DEG profiles and expression data for other transcripts are shown in Appendix A.

### 3.5. Synergistic Cell Death Induced by Upstream Activation of ER Stress

To determine whether CHOP mediated the apoptotic synergy, CHOP knockdown (siCHOP) was performed in NCI-H716 cells prior to treatment. CHOP silencing markedly attenuated growth inhibition with CBD + AZD4547, as shown by the proliferation assays (Figure 5A). Western blot analysis confirmed reduced cleaved PARP levels in siCHOP-transfected cells under combination treatment compared to control siRNA cells (Figure 5B). These findings indicate that CHOP was a critical mediator of ER stress-induced apoptosis in FGFR2-high CRC cells following combination treatment.

## 4. Discussion

### 4.1. Comprehensive Cancer Treatment Strategies

The treatment of unresectable metastatic colorectal cancer (CRC) incorporates a comprehensive range of diverse strategies aimed at extending patient survival and improving quality of life. These strategies include chemotherapy, targeted therapy, immunotherapy, radiation therapy, and supportive care. Monoclonal antibodies targeting the Epidermal Growth Factor Receptor (EGFR) represent one of the first targeted therapies to demonstrate efficacy in CRC, both as a monotherapy and in combination with cytotoxic chemotherapy [19].

### 4.2. Advances in CRC Molecular Understanding

Next-generation sequencing (NGS) has expanded our understanding of the molecular heterogeneity in CRC, identifying potentially actionable genomic alterations in patient subgroups. Targeted therapies and immune checkpoint inhibitors have changed the therapeutic landscape, yet drug resistance frequently emerges, limiting durable responses. The limited repertoire of effective targeted options highlights the urgent need to explore new molecular targets and rational drug combinations [6,19].

### 4.3. FGFR’s Role and Limitations in CRC Treatment

FGFRs are receptor tyrosine kinases implicated in development, angiogenesis, and oncogenesis. Genetic alterations in FGFRs are detected in ~7% of cancers, and in ~4–5% of CRC cases [10]. Although some FGFR-targeted therapies are approved for other malignancies, their clinical benefit in CRC has been modest, in part due to the low prevalence of FGFR alterations and gastrointestinal side effects, among other reasons [9,12,17]. Regorafenib, a multikinase inhibitor with FGFR inhibitory activity, has received FDA approval for patients with previously treated metastatic CRC. Although its approval is not based on FGFR mutation status, it may be considered for CRC patients harboring FGFR alterations when access to selective FGFR inhibitors via clinical trials is not feasible [19,20]. Considering the critical role of FGF/FGFR signaling in cell and tissue development, as well as its contribution to tumor progression, metastasis, and chemoresistance [11], FGFR remains an attractive therapeutic target. FGFR2 dysregulation in cancer can occur via amplification, mutation, or protein overexpression, which have distinct biological and clinical implications. FGFR2, in particular, has been shown to be overexpressed in CRC and is associated with tumor progression and poor prognosis [21,22,23]. In this study, we specifically investigated FGFR2 protein overexpression in the NCI-H716 CRC cell line by Western blotting, without direct assessment of gene amplification [24].

### 4.4. CBD’s Emerging Role in CRC Treatment

CBD shows promise as an adjunct therapy in CRC by impacting several key molecular pathways. It alters the tumor microenvironment to inhibit cancer progression and induces cell cycle arrest and apoptosis via a CB2-dependent mechanism [25,26]. Furthermore, CBD activates the MAPK pathways, which are critical for initiating apoptosis, paraptosis, and autophagy in CRC cells [17,27]. The efficacy of CBD in promoting apoptosis is enhanced by its ability to increase mitochondrial ROS and trigger ER stress, leading to the activation of apoptosis-promoting proteins like Noxa [15,16]. These pleiotropic effects make CBD an attractive candidate for combination therapy.

### 4.5. Rationale for Combining CBD with FGFR Inhibition

While CBD alone can induce ER stress-mediated apoptosis, FGFR signaling has been implicated in suppressing ER stress-induced cell death, partly through FGF2-mediated downregulation of CHOP via the ERK1/2 pathway [18,28]. Thus, inhibiting FGFR may release this suppression, potentially amplifying CBD-induced ER stress and apoptosis. Our RNA-seq data revealed that CBD + AZD4547 combination treatment significantly upregulated ER stress-related genes, particularly ATF3 and CHOP, supporting this hypothesis.

### 4.6. Mechanistic Findings from This Study

The CBD + AZD4547 combination synergistically induced apoptosis in FGFR2-high CRC cells (NCI-H716) but not in FGFR2-low lines. RNA-seq and Western blot analyses confirmed marked upregulation of ER stress–-elated genes and proteins. siRNA-mediated CHOP knockdown attenuated apoptosis, validating CHOP’s central role. These findings suggest that FGFR inhibition potentiates CBD’s pro-apoptotic effects via enhanced ER stress signaling.

### 4.7. Limitations and Future Directions

While this preclinical study presents promising mechanistic findings, several limitations should be noted. First, the in vitro design could not fully recapitulate the complex tumor microenvironment; thus, preclinical validation in appropriate animal models is needed to assess the anti-tumor efficacy, pharmacokinetics, toxicity, and drug–drug interactions of the CBD + FGFR inhibitor combination before early phase clinical trials. Second, although CHOP knockdown supported its role in mediating apoptotic synergy, CHOP overexpression rescue experiments were not performed, representing a mechanistic limitation. While other pathways such as ROS generation, MAPK activation, and autophagy were not extensively analyzed, RNA-seq results indicated that ER stress signaling was the most significantly enriched pathway; therefore, the mechanistic investigations in this study were focused on validating this axis. Third, the pharmacokinetics and bioavailability of CBD—affected by first-pass metabolism, lipophilicity, and formulation—should be optimized to ensure consistent therapeutic levels in vivo. Fourth, as this work examined a limited number of CRC cell lines, broader validation across diverse CRC subtypes and patient-derived models is warranted. Finally, the relatively low frequency of FGFR alterations (~4–5% in CRC) may limit applicability; therefore, identifying the molecular subgroups most likely to benefit from FGFR inhibition will be critical for clinical translation.

## 5. Conclusions

In conclusion, the data from this preclinical study indicate that the combination of cannabidiol (CBD) and FGFR inhibitors such as AZD4547 represents a potential therapeutic approach for metastatic colorectal cancer (CRC). This synergistic effect could help address resistance mechanisms that currently limit the efficacy of anticancer drugs. Our findings also suggest that ER stress-mediated apoptosis may be an important mechanism underlying this synergy. While these results are encouraging, further validation in appropriate preclinical animal models and, ultimately, clinical studies will be essential to confirm efficacy, assess safety, and determine the translational applicability of this combination strategy.

## Figures and Tables

**Figure 1 nutrients-17-02609-f001:**
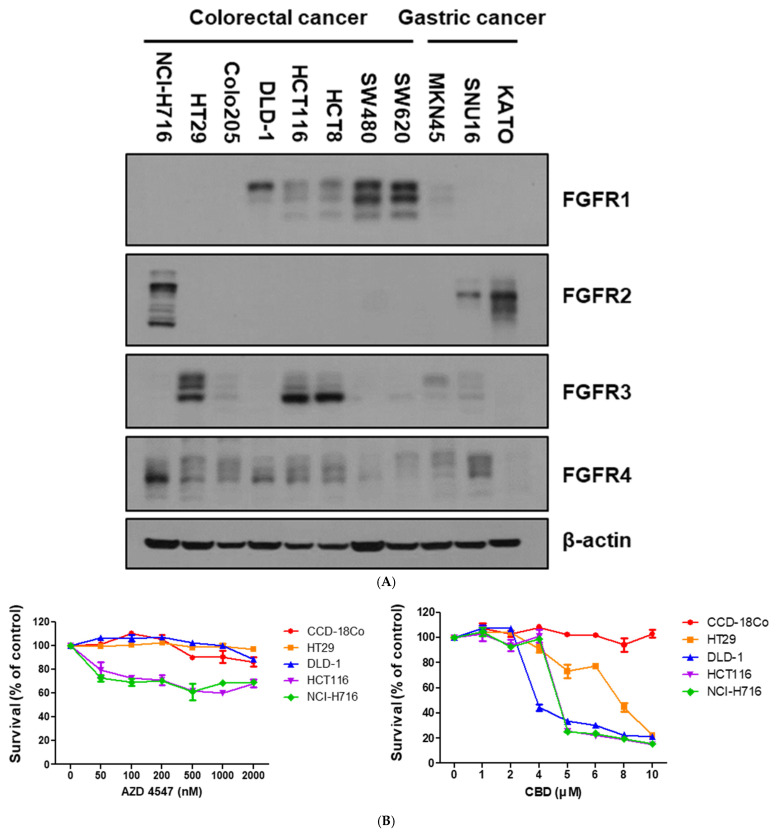
Confirmation of FGFR expression, and evaluation of FGFR inhibitor and CBD effects in CRC lines. (**A**) Endogenous protein levels of FGFR family in various cancer cell lines were determined by Western blot analysis. β-actin was used as a loading control. (**B**) Colorectal cancer and normal colon cell lines were treated with AZD4546 and CBD for 24 h. Cell viability was measured by WST-1 assay. (**C**) NCI-H716 cells were treated with increasing concentrations of CBD and AZD4547 for 24 h, and cell proliferation was measured by WST-1 assay. (**D**) Western blot analysis of NCI-H716 cells treated with AZD4547 (10 nM) for 24 h, showing inhibition of FGFR downstream signaling pathways. β-actin was used as a loading control.

**Figure 2 nutrients-17-02609-f002:**
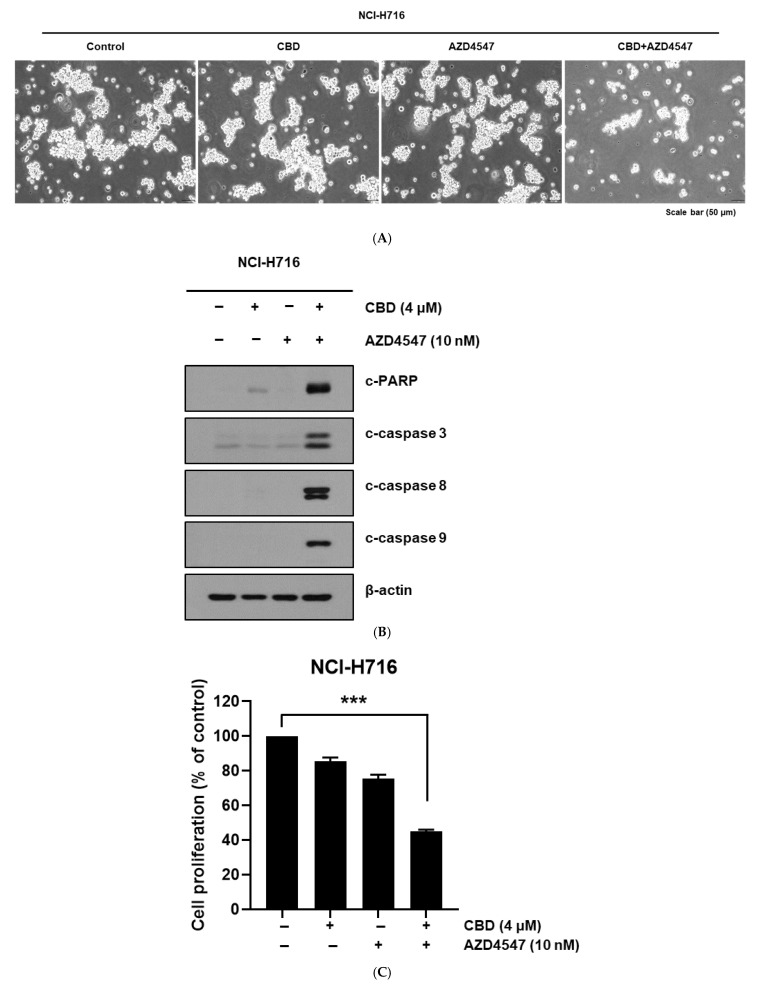
Morphological and molecular changes in NCI-H716 cells after treatment with CBD and AZD4547. (**A**) Representative phase-contrast microscopy images showing morphological changes in NCI-H716 cells following 24 h treatment with vehicle control, CBD (4 μM), AZD4547 (10 nM), or their combination. Scale bar = 50 μm. (**B**) Western blot analysis of apoptosis-related proteins in NCI-H716 cells treated for 24 h with CBD (4 μM), AZD4547 (10 nM), or their combination. Cleaved PARP, cleaved caspase-3, and cleaved caspase-8 were examined. β-actin was used as a loading control. (**C**) Quantification of apoptotic cell death determined by flow cytometry analysis using Annexin V-FITC/PI staining after 24 h treatment with CBD (4 μM), AZD4547 (10 nM), or their combination. Data are presented as means ± SD; *** *p* < 0.001 compared with single-agent treatment.

**Figure 3 nutrients-17-02609-f003:**
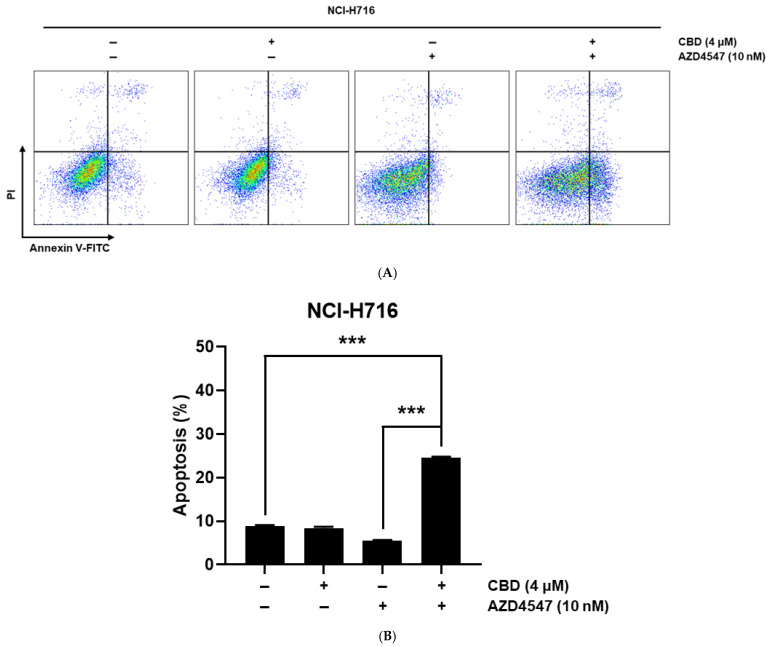
Apoptosis analysis of NCI-H716 cells following treatment with CBD and AZD4547. (**A**) Flow cytometry analysis of apoptotic cell death in NCI-H716 cells following treatment with CBD and AZD4547. Apoptotic cells were detected using Annexin V-FITC and PI staining. (**B**) Quantification of apoptotic cell death based on Annexin V/PI staining in NCI-H716 cells treated with CBD, AZD4547, or their combination. The bar graph represents the percentage of apoptotic death cells. Data are shown as the mean ± SD and analyzed by unpaired *t* test. *** *p* < 0.001.

**Figure 4 nutrients-17-02609-f004:**
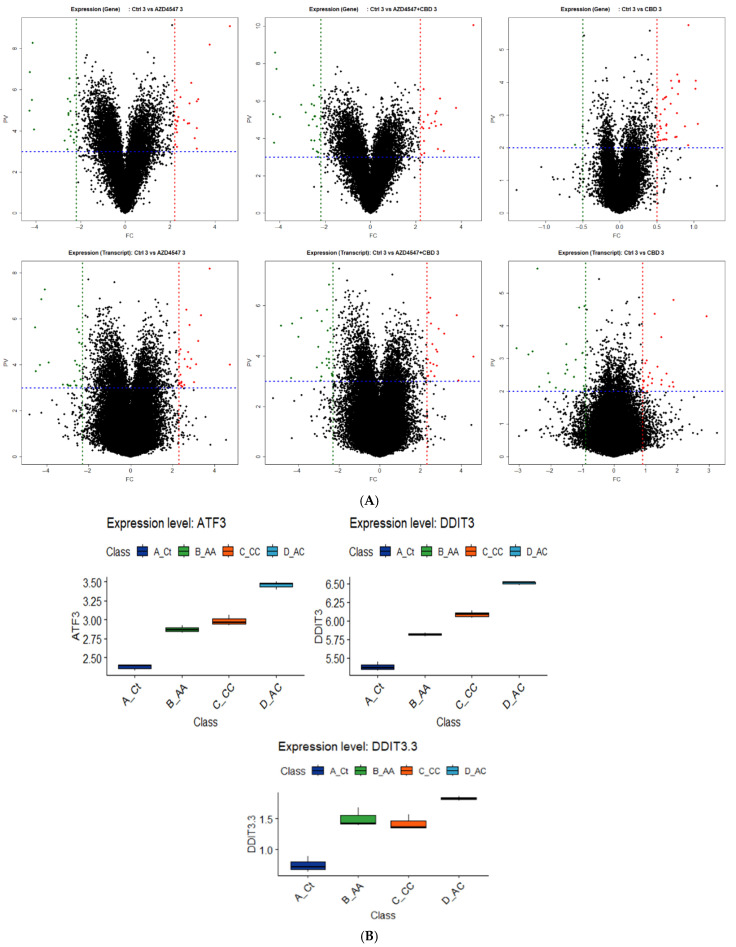
Transcriptomic changes induced by AZD4547, CBD, and their combination in NCI-H716 cells. (**A**) Volcano plots showing differentially expressed genes and transcripts in NCI-H716 cells treated with AZD4547 (10 nM), CBD (4 μM), or the combination for 24 h compared with control. In the volcano plots, red dots represent significantly upregulated genes/transcripts, green dots represent significantly downregulated genes/transcripts, and black dots represent non-significant genes/transcripts. (**B**) Box plots showing normalized expression of key ER stress-related genes from gene-level expression analysis (ATF3, DDIT3) and from transcript-level expression analysis (DDIT3.3), all of which were significantly upregulated by the combination treatment.

**Figure 5 nutrients-17-02609-f005:**
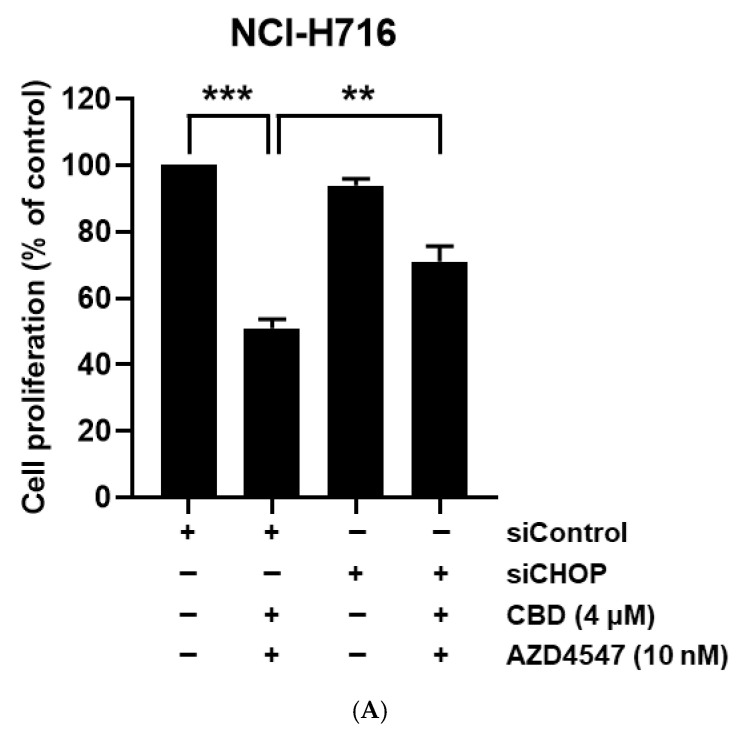
Validation of CHOP involvement in combination treatment-induced apoptosis. (**A**) Cell proliferation assay showing the effect of CHOP (DDIT3) knockdown by siRNA (siCHOP) in NCI-H716 cells. ** *p* < 0.01, *** *p* < 0.001 compared with siControl cells under the same treatment condition. The + and − signs indicate the presence (+) or absence (−) of each treatment. (**B**) Western blot analysis showing changes in CHOP and cleaved PARP (c-PARP) levels after siCHOP transfection and subsequent treatment with AZD4547 (10 nM), CBD (4 μM), or their combination for 24 h. β-actin served as a loading control.

## Data Availability

The data presented in this study are available from the corresponding author upon reasonable request. RNA sequencing data and other supporting datasets may be deposited in a public repository following publication.

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
