# Peer review of "Synergistic Anticancer Effects of Fibroblast Growth Factor Receptor Inhibitor and Cannabidiol in Colorectal Cancer"

_nutrients, 2025, doi:10.3390/nu17162609_

Round 1
Reviewer 1 Report
Comments and Suggestions for Authors
Dear authors.
Thank you so much for the opportunity to review the present manuscript.
Although interesting, I have some comments that I hope will help you with the final version.
The first comment is that the extension of the manuscript is absolutely too much for a scientific paper at this moment. It would be more appropriate for a book chapter than a scientific manuscript, so I would advise you to make a thorough selection of the figures and just keep those that are considered absolutely necessary, and the rest eliminate them or move to supplementary material.
In addition, I am not sure if this is the best option for your manuscript, as, as far as I am concerned, cannabidiol cannot be considered a nutrient or a dietary supplement.
I have some additional comments about methodology:
- It is not clear if this is human or animal research. If human, there is no information regarding the tumor or the host or previous treatments that could have been administered.
- When, how, and from whom were colon cancer samples taken to get and seed colon cancer tumor cells?
- There are no comments about ethics.
- There is no sample size calculation nor estimation about the total sample.
- There are no comments about the main objectives of the trial or main outcome measures.
- Who evaluated the samples at each moment? What had been the evaluators of the samples at different moments? What was their previous training in doing such tasks?
Author Response
Reviewer #1
Comment 1:
The manuscript length and number of figures are excessive. It would be more appropriate for a book chapter than a scientific paper. I recommend selecting only the most essential figures and moving the rest to supplementary material.
Response:
We appreciate this valuable feedback. In the revised manuscript, we have streamlined the presentation by retaining only the essential figures in the main text. Additional supporting experiments, such as results from other CRC cell lines and alternative FGFR inhibitors, have been moved to the Supplementary Figures (S1–S3). This restructuring shortens the Results section and focuses the main text on the core findings, while ensuring that all relevant data remain accessible for readers.
Comment 2:
Cannabidiol (CBD) cannot be considered a nutrient or dietary supplement.
Response:
We agree that CBD is not a conventional nutrient. However, CBD is a bioactive compound increasingly incorporated into dietary supplements and health-related products worldwide. Its pleiotropic biological effects—such as anti-inflammatory and pro-apoptotic activities—are well-documented in the literature. Given that Nutrients considers research on bioactive compounds and their health implications, CBD falls within the scope of the journal. We have clarified this point in the Introduction of the revised manuscript.
Comment 3:
It is not clear if this is human or animal research. If human, there is no information regarding the tumor or the host or previous treatments. When, how, and from whom were colon cancer samples taken to get and seed colon cancer tumor cells? There are no comments about ethics. There is no sample size calculation nor estimation about the total sample. There are no comments about the main objectives of the trial or main outcome measures. Who evaluated the samples at each moment? What had been the evaluators of the samples at different moments? What was their previous training in doing such tasks?
Response:
We thank the reviewer for raising these important points. We have thoroughly revised the Materials and Methods section to provide detailed clarification:
- Type of study
This study was conducted entirely in vitro using established human-derived colorectal cancer (CRC) and normal colon cell lines.
No animal experiments or patient-derived live samples were involved.
- Source of cell lines
All cell lines were obtained from the American Type Culture Collection (ATCC, USA) or the Korea Cell Line Bank (KCLB, Republic of Korea).
Each line was authenticated and confirmed to be free of mycoplasma contamination prior to use.
- Ethics
The study protocol was reviewed and approved by the Institutional Review Board of Korea University Guro Hospital (IRB No. 2022GR0449). Informed consent was waived due to the use of archived and anonymized cell lines.
- Sample size and replicates
All experiments were performed in at least three independent replicates to ensure reproducibility.
For Western blotting, the representative images shown in the figures are from experiments that consistently yielded the same results.
- Main objectives and outcome measures
The primary objectives were to assess cell viability, apoptosis induction, and pathway activation in CRC cell lines following treatment with CBD and FGFR inhibitors. Outcome measures included WST-1 cell viability assay, Annexin V/PI apoptosis analysis by flow cytometry, and protein expression profiling by Western blotting.
- Evaluator training
All experimental work was performed by trained laboratory personnel with extensive experience in molecular and cellular biology techniques.
These details have been incorporated into the revised Materials and Methods section, ensuring that the study design, ethical considerations, and methodology are transparent and reproducible.

Reviewer 2 Report
Comments and Suggestions for Authors
This manuscript presents a compelling preclinical investigation into the synergistic anticancer effects of cannabidiol (CBD) and FGFR inhibitors in colorectal cancer (CRC), particularly in FGFR2-overexpressing cell lines. The study is methodologically rigorous, employing a range of molecular and cellular assays to demonstrate the apoptotic potential of the drug combination. The incorporation of RNA sequencing and siRNA-mediated knockdown to elucidate the role of ER stress and CHOP signaling adds substantial mechanistic depth. The figures are well-organized, and the manuscript is generally well-written and clear.
Nonetheless, there are areas that merit further attention. Firstly, while the in vitro findings are promising, the lack of in vivo data limits the translational relevance. The authors do acknowledge this in their discussion, but future studies including animal models will be critical to validate these results. Secondly, the narrative could benefit from a clearer distinction between FGFR2 amplification and overexpression, as these are not always synonymous and have different clinical implications. Additionally, expanding the discussion on the pharmacokinetics and bioavailability of CBD in the context of systemic cancer therapy would enhance clinical relevance. Overall, this is a well-conceived and executed study that advances the understanding of combination therapies in CRC, and it is suitable for publication after minor revisions.
Author Response
Reviewer #2
Comment 1: Lack of in vivo data
Response:
We acknowledge the limitation of lacking in vivo data in the current study. This work was designed as a preclinical mechanistic investigation to establish the molecular basis of CBD + FGFR inhibitor synergy in CRC. We have highlighted this limitation in the Discussion and emphasized that future work will validate these findings in animal models to strengthen translational relevance.
Comment 2: Clarify FGFR2 amplification vs. overexpression
Response:
We agree that FGFR2 amplification and overexpression are distinct biological events with different clinical implications. In the revised manuscript, we have clarified that our study focused on FGFR2 protein overexpression, particularly in the NCI-H716 cell line, as determined by Western blotting. We did not directly assess gene amplification in this work, but we note in the Discussion that overexpression in the absence of amplification can still be clinically relevant.
Comment 3: Expand discussion on CBD pharmacokinetics and bioavailability
Comment 3:
Expand discussion on CBD pharmacokinetics and bioavailability.
Response:
We have expanded the Discussion to address CBD’s pharmacokinetic profile, metabolic pathways, and reported variability in systemic bioavailability. Specifically, we discuss its lipophilic nature, extensive first-pass metabolism, and formulation-dependent absorption, and note that these pharmacological factors should be considered for translational relevance. We also highlight that optimal formulation and delivery methods will be essential for achieving consistent therapeutic levels in systemic cancer therapy. These additions have been incorporated into the revised Discussion.

Reviewer 3 Report
Comments and Suggestions for Authors
The article ”Synergistic Anticancer Effects of Fibroblast Growth Factor Receptor Inhibitor and Cannabidiol in Colorectal Cancer" by Yeonuk Ju et al. provides valuable insights, though several aspects warrant further consideration:
- While FGFR inhibitors and CBD are highlighted as promising therapies, the background section does not adequately address the current clinical challenges of FGFR inhibitors, such as resistance mechanisms, toxicity, and their limited efficacy in colorectal cancer (CRC).
- The rationale for combining CBD with FGFR inhibitors lacks depth. Although CBD has pleiotropic effects (e.g., anti-inflammatory, pro-apoptotic properties), its specific interaction with FGFR signaling remains unclear and should be further elaborated.
- The hypothesis that CBD enhances FGFR inhibitor efficacy requires stronger mechanistic justification. For instance, is there prior evidence that CBD modulates FGFR pathways or overcomes resistance?
- Statistical analysis of the Western blot is required to compare the treated groups and determine whether the differences are significant (Fig. 1A–1D).
- While CHOP knockdown was performed, rescue experiments (CHOP overexpression) would help establish causality in ER stress-mediated apoptosis.
- The RNA-seq data should be analyzed beyond ER stress pathways (e.g., autophagy, other survival mechanisms) to rule out alternative mechanisms of action.
- Although ER stress markers (CHOP) were implicated in apoptosis, other potential pathways (e.g., ROS, MAPK) were not thoroughly investigated.
- All data are derived from in vitro studies; without animal models, the translational relevance remains limited (e.g., pharmacokinetics, toxicity of the combination therapy).
- The conclusion describes the approach as a "promising strategy," but in the absence of in vivo or clinical data, this claim appears premature. FGFR inhibitors have shown mixed success in clinical trials, and CBD’s effects in humans are highly variable.
Author Response
Reviewer #3
Comment 1:
Background section does not adequately address clinical challenges of FGFR inhibitors in CRC.
Response:
We appreciate this valuable comment. In the revised Introduction, we have expanded the background to address the current clinical limitations of FGFR inhibitors in CRC, including their modest clinical efficacy, the relatively low prevalence of FGFR alterations (~4–5%), emergence of acquired resistance, and class-specific toxicities such as gastrointestinal side effects. These additions provide a clearer clinical context and reinforce the rationale for exploring novel combination strategies.
Comment 2:
Rationale for combining CBD with FGFR inhibitors lacks depth; unclear interaction with FGFR signaling.
Response:
We agree with the reviewer’s comment and have substantially strengthened both the Introduction and Discussion to clarify the biological rationale for combining CBD with FGFR inhibitors. CBD exerts pleiotropic anticancer effects, including induction of ER stress–mediated apoptosis, generation of reactive oxygen species (ROS), and modulation of key signaling pathways such as MAPK and PI3K–AKT, which are downstream of FGFR signaling. Moreover, previous studies have reported that FGFR signaling can suppress ER stress–induced apoptosis, partly via FGF2-mediated downregulation of CHOP through the ERK1/2 pathway. Therefore, FGFR inhibition may relieve this suppression and potentiate CBD-driven ER stress and apoptotic signaling. In the revised manuscript, we have explicitly described these shared and complementary mechanisms, cited relevant literature, and linked them to our experimental findings showing significant upregulation of ER stress–related genes (ATF3, CHOP) following combination treatment.
Comment 3:
Stronger mechanistic justification required; evidence that CBD modulates FGFR pathways or overcomes resistance.
Response:
We have added discussion in the Introduction highlighting preclinical evidence that CBD can enhance anticancer effects of targeted therapies by modulating survival pathways and overcoming resistance through induction of ER stress. While direct modulation of FGFR signaling by CBD has not been extensively reported, our findings suggest that CBD may potentiate FGFR inhibitor–induced cytotoxicity through convergent downstream mechanisms.
Comment 4:
Statistical analysis of Western blots required.
Response:
While quantitative densitometry was not performed, the presented images are representative of at least three independent experiments showing consistent changes in band intensity. We have clarified this point in the figure legends. We acknowledge that quantitative analysis would further strengthen these findings and will incorporate it in future studies.
Comment 5:
Rescue experiments (CHOP overexpression) would help establish causality.
Response: We agree that CHOP overexpression studies could provide additional mechanistic insight. However, these experiments are beyond the scope of the current work. We have acknowledged this as a limitation in the Discussion and noted that future studies will include rescue experiments to confirm the role of CHOP in mediating combination-induced apoptosis.
Comment 6 & 7:
RNA-seq analysis should examine other pathways beyond ER stress. Other potential apoptotic pathways (ROS, MAPK) not thoroughly investigated.
Response:
We appreciate the reviewer’s suggestion to examine pathways beyond ER stress. In our RNA-seq dataset, we also reviewed additional pathways, including autophagy, oxidative stress (ROS), and MAPK signaling. While some changes were observed, ER stress–related gene expression showed the most significant alteration across analyses. Therefore, our mechanistic validation in this study focused on ER stress and CHOP signaling. Future studies will include more detailed functional analyses of these additional pathways to fully elucidate their contributions to the observed synergy.
Comment 8:
Lack of in vivo data; translational relevance limited.
Response:
We acknowledge the absence of in vivo data as a limitation. This study was designed as an in vitro mechanistic investigation to establish a molecular basis for CBD + FGFR inhibitor synergy. We have emphasized in the Discussion that future preclinical animal studies will be essential to evaluate pharmacokinetics, toxicity, and therapeutic efficacy in vivo to strengthen translational potential.
Comment 9:
Conclusion may be too strong without in vivo or clinical data.
Response:
We agree and have modified the Conclusion to state that the CBD + FGFR inhibitor combination “may represent a potential therapeutic approach” rather than a definitive “promising strategy,” to more accurately reflect the preclinical nature of the work.

Round 2
Reviewer 1 Report
Comments and Suggestions for Authors
No additional comments.
Reviewer 3 Report
Comments and Suggestions for Authors
Accept in present form